# A DR/NIR Hybrid Polymeric Tool for Functional Bio-Coatings: Theoretical Study, Cytotoxicity, and Antimicrobial Activity

**DOI:** 10.3390/polym15040883

**Published:** 2023-02-10

**Authors:** Rosita Diana, Francesco Silvio Gentile, Simona Concilio, Antonello Petrella, Raffaella Belvedere, Martina Schibeci, Angela Arciello, Luigi Di Costanzo, Barbara Panunzi

**Affiliations:** 1Department of Agricultural Sciences, University of Naples Federico II, 80055 Napoli, Italy; 2Department of Pharmacy, University of Salerno, 84084 Salerno, Italy; 3Bionam Research Center for Biomaterials, University of Salerno, 84084 Salerno, Italy; 4Department of Chemical Sciences, University of Naples Federico II, 80126 Napoli, Italy; 5Istituto Nazionale di Biostrutture e Biosistemi (INBB), 00136 Roma, Italy

**Keywords:** hybrid polymer, biomedical, DR/NIR fluorescence, antimicrobial, zinc complex

## Abstract

Among modern biomaterials, hybrid tools containing an organic component and a metal cation are recognized as added value, and, for many advanced biomedical applications, synthetic polymers are used as thin protective/functional coatings for medical or prosthetic devices and implants. These materials require specific non-degradability, biocompatibility, antimicrobial, and antiproliferative properties to address safety aspects concerning their use in medicine. Moreover, bioimaging monitoring of the biomedical device and/or implant through biological tissues is a desirable ability. This article reports a novel hybrid metallopolymer obtained by grafting zinc-coordinated fragments to an organic polymeric matrix. This hybrid polymer, owing to its relevant emission in the deep red to near-infrared (DR/NIR) region, is monitorable; therefore, it represents a potential material for biomedical coating. Furthermore, it shows good biocompatibility and adhesion properties and excellent stability in slightly acidic/basic water solutions. Finally, in contact with the superficial layers of human skin, it shows antimicrobial properties against *Staphylococcus aureus* bacterial strains.

## 1. Introduction

Modern biomaterials are biocompatible tools aiming for a minimal host response for applications in medicine and research involving living organisms [1,2]. The research in biomaterials for applications in specific parts of the human body has become a hotspot for the materials industry of the XX century as the growing biomedical technology deals with tissue engineering, controlled drug release, and implants [3,4]. Nowadays, modern synthetic biomaterials such as metal alloys, ceramics, and synthetic polymers are frequently placed alongside natural polymers such as collagen, polysaccharides, and gelatins [3]. Synthetic organic polymers such as silicones, polyvinylchloride, polyurethanes, and polycarbonates have gained widespread attention for biomedical implants and coating devices and tools due to the ability to tailor the structure to meet specific applicative requirements [5]. Synthetic polymers containing an inorganic component are playing a revolutionary role in biomedical science [3,6,7]. The added value of these hybrid materials is represented by their advanced and tunable properties due to the versatility of the organic parts and their cooperative ability with some metals to form a variety of mono- or poly-dimensional structures. Versatile modulation of chemical, thermal, electrical, mechanical, and spectroscopic properties and functions is also expected for a usable hybrid material. A device/implant for use in the living body must meet targeted requirements of toxicity, degradability, and biocompatibility depending on the tissue and organ in which it is implanted. Finally, their properties need to meet the requirements of the specifically employed technology [1,8]. Synthetic biopolymers are often used as an interface between two materials or as the coadjuvant material for the device/implant. In many advanced biomedical applications, the polymers are not present as bulk materials but as thin protective/functional coatings for medical and prosthetic devices and implants [5,9,10,11,12,13,14].

Biopolymers used as biomedical coatings must be biocompatible with their contact surface counterparts, which should benefit from their antimicrobial and antiproliferative properties [3,9,15,16].

Another critical issue in biomedical technology is monitoring the biomedical device and/or implant [17,18,19,20,21,22,23]. To determine and monitor a device’s location, such as a stent or a catheter, clinicians typically utilize X-ray imaging. Despite this well-recognized standard technique, prolonged ionizing radiation exposure in X-ray imaging represents a risk. An attractive alternative is the near-infrared (NIR) imaging technique, which allows imaging without harmful side effects [24,25,26,27,28]. The semitransparency of biological tissue in the range of 700–1000 nm makes NIR emission particularly advantageous for any biomedical optical probes. The main tissue components have absorption bands below 600 nm wavelength, while water absorbs above 1150 nm, so the bioimaging technique can be employed in the window between 650 and 950 nm (deep red/near-infrared, DR/NIR). This guarantees deep tissue penetration, minimum biological photodamage, and minimum interference from the background autofluorescence of the living tissue components [29]. 

This article reports the design and synthesis of a novel DR/NIR fluorescent hybrid metallopolymer (named P4VP-ZnNB, see Figure 1) as a potential biomedical material. The P4VP-ZnNB polymer was obtained by integrating a DR/NIR fluorophore (ZnNB in Figure 1) contained as a low percentage within a polymeric organic matrix. The fluorogen unit was obtained from a (hydroxynaphthalen)-benzodifuran ligand (NB) in Figure 1) acting as a chelator toward the zinc (II) cation and known for its biological properties [30]. The polymeric grafting matrix is poly(4-vinylpyridine) (P4VP). Besides its biocompatibility [31], such a versatile polymer is suitable for several applications, such as pH-sensitive systems, antibacterial surfaces, the immobilization of nanoparticles, and functional coatings for biomedical and medical purposes [11,22,31,32,33,34,35,36,37,38]. Moreover, its ability was proven [39,40] by emphasizing the emission pattern of zinc-chelate fragments upon coordination with the donor pyridinic nitrogen atom groups. By correlation of experimental data and theoretical analysis, we provided a consistent theory with the observed fluorochromism involving the pivotal role of the zinc coordination environment.

The comprehensive exploration of the chemical and physical properties, optical behavior, and biological assays proved that the P4VP-ZnNB polymer containing 20% *w*/*w* active fluorophore is a good DR/NIR emitter with relevant photoluminescence quantum yield (PLQY) in the DR/NIR spectral region. The grafted polymer proved its stability in air, in daylight, and in contact with aqueous solutions in the pH range from 5.0 to 8.0. Furthermore, the polymer showed good biocompatibility with human keratinocytes, endothelial cells, and fibroblasts, and antimicrobial activity against *S. aureus* ATCC 29213 and the methicillin-resistant *S. aureus* WKZ-2.

## 2. Experimental Section

### 2.1. Materials

Commercially available starting products were supplied by Sigma Aldrich (20151 Milano (MI), Italy). Supplied P4VP had an average molecular weight of 60,000 Da (T_g_ = 137 °C). Compound NB was obtained as described in [30]. ^1^H NMR spectra were recorded in 1,1,2,2-tetrachlorethane-d2 with a Bruker Advance II 400 MHz apparatus by Bruker Corporation, Billerica, MA, USA. Mass spectrometry measurements were carried out using a Q-TOF premier instrument (Waters, Milford, MA, USA) with an electrospray ion source and a hybrid quadrupole-time of flight analyzer. Optical observations were performed using a Zeiss Axioscop polarizing microscope by Carl Zeiss, Oberkochen, Germany, equipped with an FP90 Mettler microfurnace by Mettler-Toledo International INC MTD, Columbus, OH, USA. The decomposition temperatures (measured as 5 wt.% weight loss) and phase transition temperatures and enthalpies were measured under nitrogen flow using a DSC/TGA Perkin Elmer TGA 4000 by PerkinElmer, Inc., Waltham, MA, USA, at scanning rate 10 °C min^−1^. Absorption and UV–visible emission spectra were recorded using the JASCO F-530 and FP-750 spectrometers by JASCO Inc., Easton, MD, USA, at a scanning rate of 200 nm min^−1^, and on a spectrofluorometer, the Jasco FP-750 by JASCO Inc., Easton, MD, USA (excitation wavelengths set at the absorption maxima of the samples), at scanning rate 125 nm min^−1^. Thin films of the samples dissolved in chloroform (approximately 10 mg/mL) were prepared using an SCS P6700 spin coater operating at 600 rpm for 1 min. The films were dried at 50 °C for 30 min and kept at room temperature for 2 h. Photoluminescence quantum efficiency values were recorded on quartz slides using a Fluorolog 3 spectrofluorometer by Horiba Jobin Instruments, SA, within an integrating sphere provided by an optical fiber connection. Morphological characterization of P4VP-ZnNB was performed with a Scanning Electron Microscope (SEM), the Leo 1530 Gemini by Zeiss, on the spun films; the operating voltage was 6 kV for all the measurements performed.

For all biological assays, coated and uncoated glass were sterilized through exposure to UV rays emitted by a lamp (36 J/s) for 15 min at room temperature. Such sterilization conditions were not harmful to the films, which showed the same spectroscopic pattern after irradiation.

### 2.2. Synthesis of ZnNB

The synthesis of the model compound was attempted both with zinc (II) acetate and zinc (II) chloride salt (in the presence of sodium acetate with the same results. For synthetic convenience, the isolated ZnNB complex obtained from zinc chloride was used for the grafting route on P4VP (see synthesis below) and for further characterizations. An amount of 0.762 g (1.00 mmol) of NB dissolved at 50 °C in 25 mL of chloroform was added to 0.082 g (1.00 mmol) of sodium acetate dissolved in a minimum amount of ethanol and 0.136 g (1.00 mmol) of zinc (II) chloride under stirring. After 15 min at boiling temperature, the crude product precipitated. The compound was recovered from the cooled solution and washed in hexane, and then in ethanol twice and dried at 40 °C overnight. Yield = 55%. T_m_ = 320 °C; T_d_ = 330 °C. ^1^H NMR (500 MHz, 1,1,2,2-tetrachlorethane (TCE)-d2, 25 °C, ppm): 1.20 (t, 3H), 1.33 (t, 3H), 3.04 (q, 2H), 3.81 (q, 2H), 6.73 (d, 1H), 6.91 (d, 1H), 7.04 (d, 1H), 7.15 (d, 2H), from 7.43 to 7.79 (m, 7H), 7.81 (d, 1H), 7.95 (dd, 1H), 8.07 (s, 1H), 8.33 (d, 1H), 8.35 (d, 1H), 9.64 (s, 1H), 9.80 (s, 1H), 14.94 (s, 1H); see hard copy in Appendix A. Elemental analysis (%) calculated for C_44_H_30_N_3_O_10_ClZn: found: C, 61.39; H, 3.58; N, 4.86, Zn 7.50; calculated: C, 61.34; H, 3.51; N, 4.88; Zn, 7.59. Zn % found (estimated by TGA analysis as ZnO): 9.40%; calculated: 9.44%. l_abs_ (crystals) = 530 nm, see the spectrum in Appendix A. Q-TOF of ZnNB m/z: 860.11 (M + H).

### 2.3. Synthesis of the Grafted Polymers P4VP-ZnNB

The grafted polymers were obtained both by in situ formation of the complex and using a preformed ZnNB complex. The grafted samples used for further characterizations were obtained by using 10%, 20%, and 30% *w*/*w* preformed ZnNB complex. The 20% sample was obtained by the following procedure. To 0.800 g of P4VP (molecular unit: C_7_H_7_N, 105.14) dissolved at 50 °C in 30 mL of chloroform, 0.200 g (0.23 mmol) of ZnNB was added under stirring. After 30 min at boiling temperature, the crude product was precipitated by pouring in hexane/ethanol (1:1) at room temperature. The compound was washed in ethanol twice and dried at 60 °C overnight. Yield was virtually quantitative. T_g_ = 154 °C; T_d_ = 340 °C. The hard copy of the ^1^H NMR spectrum, recorded by a 500 MHz, in 1,1,2,2-tetrachlorethane (TCE)-d2, 25 °C, is reported in Appendix A. Zn% found (estimated by TGA analysis as ZnO, see Appendix A): 1.88%; calculated: 1.89%. l_abs_ (amorphous film) = 520 nm, see absorbance spectrum in Appendix A. Elemental analysis (%) calculated for ZnNB 20% grafted on P4VP: found: C, 76.31; H, 6.10; N, 11.67, Zn 1.44; calculated: C, 76.06; H, 6.06; N, 11.63; Zn, 1.50.

### 2.4. Single-Crystal X-ray Analysis

Different experiments were performed at room temperature to obtain single crystals of ZnNB complex by slow evaporation. Red crystals of ZnNB appeared as small plates with typical dimensions of 0.05 × 0.02 × 0.3 mm from ethanol/water (10:1) at pH > 8. Data of ZnNB were collected with synchrotron radiation (wavelength, 0.619 Å) from the XRD1 beamline at the Elettra Synchrotron Light Source, Trieste, Italy. The energy spectra obtained from the X-ray fluorescence measurement of a single crystal and ZnNB powder showed characteristic K edge (8.6 keV) and K peaks of zinc (Appendix A). Selected crystals for data diffraction were dipped in cryoprotectant Paratone oil using a small loop of fine rayon fiber, and flash-frozen in a stream of nitrogen at 100 K. Data were processed using XDS and POINTLESS 1.11.21, with data collection statistics reported in Appendix A [41,42]. No data twinning was detected. Crystals presented a centric monocline unit cell with axes a = 16.442 Å, b = 4.790 Å, c = 25.880 Å; β = 91.24 (°) and space group *P 1 21/n 1*. The molecule’s structure was solved by direct methods using SHELXS [43], which revealed the expected skeleton corresponding to a T-shaped pattern recognizable in the two different segments: the main plane consisting of the bis(hydroxynaphthalen)-benzodifuran moiety and the nitrophenyl group. During the last stages of refinement, a residual Fourier density using coefficients |F_obs_|–|F_calc_| and a cutoff of 4σ found at a coordinative distance from oxygen or nitrogen atoms were interpreted as consistent with the low occupancy of zinc ions. However, the centrosymmetric space group did not allow for anomalous signal calculation from zinc and the unique identification of the metal ion. Therefore, we did not proceed further with refinement. Nonetheless, the unit cell parameters of the ligand crystals grown in the presence of Zn acetate were isomorphic with the monomeric form of the ligand itself, grown in the absence of metal and previously determined, and provided insights into the equimolar ratio between the NB ligand and zinc [44].

### 2.5. Cell Cultures

The HaCaT cell line (human immortalized keratinocytes) was purchased from CLS Cell Lines Service GmbH, 69214 Eppelheim, Germany. It was maintained in Dulbecco’s modified Eagle’s medium (DMEM) with 10% fetal bovine serum (FBS), as reported in [45]. The BJ cell line (human immortalized fibroblasts) was purchased from the American Type Culture Collection (ATCC^®^ CRL2522™), University Boulevard Manassas, Virginia, United States and cultured in Eagle’s Minimum Essential Medium (MEM) with 10% FBS, 1% L-glutamine, 1% sodium pyruvate, 1% NEAA. These media were supplemented with antibiotics (10,000 U/mL penicillin and 10 mg/mL streptomycin). The Human Umbilical Vein Endothelial Cells (HUVEC) cell line, cultured until passage 10 as reported in [46] (ATCC^®^ PCS-100-010™), was maintained in endothelial growth medium (EGM-2) containing EBM-2 medium (serum-free, growth-factor free), supplemented with 2% fetal bovine serum (FBS), human fibroblast growth factor-B (hFGF-B), human epidermal growth factor (hEGF), human vascular endothelial cell growth factor (hVEGF), long R insulin-like growth factor-1 (R3-IGF-1), ascorbic acid, hydrocortisone, and heparin (Lonza). Cells were stained at 37 °C in a 5% CO_2_–95% air humidified atmosphere and were serially passed at 70–80% confluence.

### 2.6. Hemocytometer Counting

HaCaT, HUVEC, and BJ cells were seeded in a 24-well-plate in a quantity of 8 × 10^4^ cells/well on 12 mm diameter coverslips (Thermo Fisher Scientific, Waltham, MA, USA) spin-coated with the polymer. The coverslips were sterilized through UV rays. After 72 h, brightfield images were captured using the EVOS optical microscope (10×) (Life Technologies Corporation, Carlsbad, CA, USA). Next, cell counting was performed as reported in [47]. The cell suspension and equal volumes of 0.4% trypan blue (Sigma-Aldrich) stain were mixed. Then, 10 μL of trypan blue/cell mix was placed at the edge of the coverslip of the Bruker chamber, and the hemocytometer grid was visualized under the optical microscope Axiovert 40 CFL (Carl Zeiss MicroImaging GmbH). To calculate the viable cells/mL, the average number of cells in one large square was multiplied by the dilution factor (2) and then by 10^4^.

### 2.7. Bacterial Strains and Growth Conditions

Bacterial strains *Staphylococcus aureus* ATCC 29213 (*S. aureus* ATCC 29213) and the clinically isolated methicillin-resistant *Staphylococcus aureus* WKZ-2 (MRSA WKZ-2) were grown in Muller–Hinton Broth (MHB, Becton Dickinson Difco, Franklin Lakes, NJ, USA) at 37 °C in shaking conditions. Colonies counting analyses were performed using Luria–Bertani agar plates (LB; tryptone 10 g; yeast extract 10 g; sodium chloride 5 g). 

### 2.8. Evaluation of Antimicrobial Activity

Bacterial cells were inoculated and grown in MHB at 37 °C to mid-logarithmic phase to test the antimicrobial activity of P4VP-ZnNB films. Afterwards, 103 CFU/cm^2^ were spotted on glass coverslips coated with P4VP alone or in the presence of 10% or 20% ZnNB. The uncoated glass was tested as a control. A 1 cm^2^ glass was placed upon the spotted culture before incubating the samples overnight at 37 °C in a humidified chamber to encourage the contact of the bacteria with the surface. After the incubation, the cells were mechanically detached by vigorous washing in 1 mL MHB; samples were then serially diluted, plated on LB agar, and incubated overnight at 37 °C to perform colony counting. The results of the assay refer to at least two independent experiments. Statistical analysis was performed by using Student’s *t*-Test. Significant differences are indicated as * (*p* < 0.05), ** (*p* < 0.01), or *** (*p* < 0.001).

### 2.9. Computational Methods

Quantum mechanical ab initio simulations were performed using the ORCA 5.0.3 package adopting a Density Functional Theory (DFT) formalism. Molecular models were sketched for NB (in acidic conditions with both hydroxyl groups protonated), ZnNB, and ZnNB(Py)_2_ (ZnNB with two coordinated pyridine molecules) and pre-optimized using Avogadro software’s version 1.2.0 molecular mechanics tools [48]. The DFT computational schemes are known [49,50], with the adoption of the Linear Combination of Atomic Orbitals (LCAO) and the hybrid functional (with a percentage of Hartree–Fock exchange to avoid the electrons’ self-interaction). In LCAO approximation, each molecular orbital is represented as the sum of a Gaussian Type Orbitals (GTO) basis set. In particular, a double-z basis set, def2-SVP [51], was adopted with the B3LYP global hybrid functional [52,53]. The effect of the exchange-only gradient correction [53] and the Grimme D3 empirical correction [54] was considered. The molecular structures were optimized with respect to the atomic coordinates. The vibrational properties were calculated from the analytical derivatives of the gradient to check the stability of each minimum within the harmonic approximation. The location of the real minimum of the optimized structures was confirmed by the positive values of all vibrational eigenvalues. 

Time-Dependent Density Functional Theory (TD-DFT) in Tamm–Dancof approximation [55] was calculated to simulate the UV adsorption spectra, considering the first forty excitation roots. A Lorentzian function to fit the excitation roots was adopted (FMHW = 20 nm) to reproduce the broad peak in the experimental conditions.

## 3. Results and Discussion

### 3.1. Design and Chemical–Physical Properties

The chemical structure of the coordination core (model compound ZnNB) is reported in Figure 1. The model compound was crystallized by the reaction of the ligand NB with zinc (II) acetate (see Section 2.4 and Appendix A) in aqueous ethanol. 

The model of the ZnNB complex served as a foundation for the coordination mode of NB to the zinc cation and to define the stoichiometry of the material. The ligand NB shows a bulky T-shaped pattern with two potential chelating sites in the electron-donor arms and an electron-acceptor nitrophenyl substituent, which in turn makes the two sides of the molecule sterically non-equivalent [30,57,58,59]. The more encumbered side due to the presence of a nitrophenyl group on the same side of the acetyl group cannot act as a chelating site. Conversely, the other half-Salen Schiff base site functions as a mono-negative tridentate site toward the zinc (II) cation, directing towards a 1:1 (ligand to metal) stoichiometry. 

The model compound ZnNB is a red crystalline material with an absorbance maximum at 530 nm in the solid state. The narrow bandgap drives the deep-red emission of the ligand [30,44]; the maximum of the emission band of the NB molecule redshifts by 10 nm upon zinc coordination. The ZnNB molecule in its crystalline state is a DR/NIR emitter, with an appreciable amount of NIR emission (see the hump above 700 nm, Figure 2, CIE (Commission Internationale de l’Éclairage) (0.44; 0.27) when excited at its λ_abs_ maximum. The role of the zinc (II) ion in improving PL performance is well known, mainly due to the increased rigidity of the molecular structure upon coordination, which leads to a decreased probability of nonradiative transitions by the excited states (chelation-induced enhanced fluorescence (CHEF) effect [60,61]). As expected, PLQY in the solid state increased from 18% for NB to 25% for ZnNB. The large Stokes shift (137 nm from the excitation wavelength to the first emission maximum) preserved from the reabsorption of the emitted photons guarantees high emission efficiency. 

Different low-molecular-weight and polymeric zinc-based complexes acting in the aggregation-induced emission enhancement (AIE) [60,62] mode previously attracted our attention [60]. Solid-state zinc complexes with the coordination sphere completed by one or two pyridine molecules did exhibit relevant and tunable photoluminescence (PL) performance, usually enhanced with respect to the same complexes in the absence of the pyridine ligands [63]. With the aim of transferring the photoluminescence performance of the coordination fragment ZnNB to the metallated polymer, we grafted the ZnNB complex on a non-emissive P4VP skeleton using different percentages (10%, 20%, and 30% *w*/*w*) of the complex and a 60,000 Da polymer. The 30% sample was not considered for further exploration due to its scarce solubility and processability. For both 10% and 20% samples, the emission pattern was similar (PLQYs increase from around 30% to 35%), and toxic release remained absent, while the antimicrobial responses became worse. As the best balance between PL performance and biological requirements was obtained with the 20% *w*/*w* ZnNB in P4VP (named P4VP-ZnNB 20%), it was fully characterized.

The polymer P4VP-ZnNB 20% showed quite nominal zinc content (as derived by crossing thermal and elemental analysis, see Section 2.3), indicating that the coordination fragment was fully grafted. The sample was a red amorphous material, highly transparent when processed in thin films. Its T_g_ was higher than that of neat P4VP (154 °C vs. 137 °C). SEM analysis confirmed the homogeneous non-structured nature of the spin-coated thin films. The P4VP polymer is well known for its strong adhesion to many metal and nonmetal surfaces [64], well preserved in the P4VP-ZnNB 20% polymer (Appendix A). The spin-coated films of P4VP-ZnNB 20% were resistant to redissolving with solvent and scratching, even more so than the neat P4VP matrix. This effect can be ascribed to the ability of the zinc cation to bind one or even two pyridine nitrogen groups, causing a low degree of crosslinking [39]. 

A systematic analysis of the stability in ordinary conditions of use and in solution was performed on P4VP-ZnNB 20%. A set of thin films of P4VP-ZnNB 20% spin-coated onto quartz slides were kept at room temperature and under daylight for over six months. Their habitus, SEM pattern, and absorbance/emission spectroscopic pattern remained unaltered. Another set of P4VP-ZnNB 20% samples spin-coated onto quartz slides were immersed in aqueous solutions buffered at a slightly acidic/basic pH (respectively at 5.0, 5.5, 6.5, 7.5, 8.0 pH values) for two weeks. The films did not detach from the quartz holder. Moreover, in this case, their habitus, SEM pattern, and absorbance/emission spectroscopic pattern remained unaltered. The ^1^H-NMR spectra of the same polymeric films immersed in the buffered aqueous solutions, and dried in air, were unaltered with respect to the ^1^H-NMR spectrum of untreated P4VP-ZnNB 20%. Finally, the ^1^H-NMR spectrum of untreated P4VP-ZnNB 20% recorded before and after two weeks was unaltered.

Due to the potential as a functional coating, an in-depth spectroscopic analysis was performed on P4VP-ZnNB 20% in the solid state. The emission CIE coordinates (0.55; 0.31) placed solid P4VP-ZnNB 20% in the DR/NIR region (see Figure 2), with a Stokes shift of 101 nm from the excitation wavelength (broad absorbance band peaked at 520 nm) to the first emission maximum (621 nm). DR/NIR emitters with significant quantum yields in the solid-state are rare, even between hybrid materials [14,28,65,66,67,68,69,70], and PLQY is a crucial parameter. Despite the low percentage of fluorophore in P4VP-ZnNB 20%, the presence of a lowest unoccupied molecular orbital (LUMO) band due to many pyridinic groups enhances the PLQY up to 35%. Due to the dilution imposed by the non-emissive polymer matrix, the emission band of the ZnNB fragment broadens and undergoes a blue shift (see Figure 2, from 667 to 621 nm). Interestingly, P4VP-ZnNB 20% still retains significant NIR emission (almost a quarter of the emission band falls between 700 and 900 nm, see Figure 2), when excited at its λ_abs_ maximum. Moreover, almost one half of the emission band falls within the spectral zone typically employed in bioimaging techniques (starting from 650 nm, see Figure 2) [71,72]. Finally, as DR/NIR tools can benefit from the use of a standard excitation wavelength [73,74], we used a commercial 365 UV lamp as a cheap instrument to observe fluorescence via easy in-situ observation on P4VP-ZnNB 20% samples (Figure 2, inset on the right). 

### 3.2. Theoretical Results

The electronic modification can be represented by differential Homo–Lumo densities (ρ_diff_ = ρ_H_−ρ_L_). The gaps correspond to an alteration of the frontier electronic levels. Homo–Lumo gaps in the ground state geometries (absorption calculations, Figure 3 above) span from 2.46 to 2.26 eV, with a slight, progressive narrowing when the free ligand NB is compared with ZnNB and ZnNB(Py)_2_. The ρ_diff_ iso-level patterns are almost diffuse in the presence of pyridine-coordinated molecules, with stronger involvement of the -NO_2_ moiety and the peripheral region of the NB molecule.

Similar behavior is observable in the differential densities concerning the relaxation of the first excited state (following the Kasha rule). In emission calculations (Figure 3 below), Homo–Lumo gaps span from 2.24 to 1.76 eV. As expected, the metal is not involved in the charge-transfer process [60]. The strongly enhanced pattern identified by the red lobes positioned on the benzene rings near the metal does not involve the pyridine rings but can be ascribed to the interaction of the metal with the lone pairs of the pyridinic nitrogen atom groups.

The effects of pyridine ligands can be correlated to Homo–Lumo energy level modifications in the relaxation process from the ground state (Homo energy, E_H_) to the first excited state (Lumo energy, E_L_). As shown in Table 1, Homo destabilization (∆E_H_ = E_H_^B^−E_H_^A^) for NB, ZnNB, and ZnNB(Py)_2_ is similar. Regarding Lumo stabilization, ∆E_L_ values (∆E_L_ = E_L_^B^−E_L_^A^) significantly decrease from the unbonded ligand to the pyridinic complex. The pattern reflects the tendency of pyridine ligands to destabilize Homo (0.24 eV) and even more to stabilize Lumo (−0.30 eV), producing a remarkable energy gap reduction (1.71 eV) compared to NB and ZnNB. 

The absorption and emission spectra simulations are shown in Figure 4. Interestingly, NB and ZnNB show different shapes with respect to the ZnNB(Py)_2_ complex. The most intense absorbance peaks are very similar for NB and ZnNB, falling at 504 and 508 nm, respectively. Conversely, the absorbance for the pyridinic complex ZnNB(Py)_2_ is redshifted at 554 nm (first peak, Homo–Lumo transition). Consistent with the experimental data, theoretical predictions provide redshifted emission for NB and ZnNB, falling at 555 and 586 nm, respectively. The coordination of two pyridinic units on the coordination core strongly drives the emission to the NIR zone, with a calculated maximum at 724 nm. Obviously, the complex ZnNB(Py)_2_ cannot be directly compared with the grafted polymer, as the dilution effect due to the pyridine chains strongly affects the emission pattern. Nevertheless, it indicates an interesting trend towards NIR emission due to pyridine coordination.

### 3.3. Biological Assays

The concept of the biocompatibility of a coating layer is strictly dependent on the targeted application and technology, the time (permanent or transient), and the type (direct or indirect) of contact [5]. Therefore, we directed the biocompatibility investigations towards the first cell lines related to human skin contact.

On the other hand, naphtholic and benzodifuranic units are known for their biological activity, such as antimicrobial [75,76], antiviral [77,78], antifungal [76,79], and analgesic [80]. Complexes of divalent cations (such as zinc, nickel, and cobalt) with Schiff base ligands were obtained from 2-hydroxy-1-naphtaldehyde [81,82]. P4VP, well known as a versatile polymer for functional coatings [11], also demonstrated antimicrobial properties in its derivatives [83]. Therefore, we checked the polymer’s antimicrobial activity by addressing two ubiquitous pathogenic bacteria. 

We employed two polymer samples for the biological assays, 10% and 20% *w*/*w* P4VP-ZnNB, to compare results when increasing the active units. Based on the collected data, the overall functional performance of the 10% sample was better.

#### 3.3.1. Cytotoxicity Assay

The in vitro compatibility of the polymer with cells has been evaluated on the HaCaT, HUVEC, and BJ cell lines, as human keratinocytes, endothelial cells, and fibroblasts, respectively, to assess the macroscopic effects of P4VP-ZnNB both at 10% and 20% *w*/*w* in contact with the superficial layers of human skin. 

Figure 5A reports the representative images of cells seeded for 72 h on coverslips with or without the polymer coating. No significant differences have been highlighted when cells are in contact with the polymer compared to control experimental points, represented by non-treated coverslips. In detail, the images highlighted a very similar state of health of the treated and non-treated cells, which appeared to refract the microscope’s light, with well-defined contours denoting continuous plasma membranes without any breakage. Furthermore, we observed the absence of granularity around the nucleus and of vacuoles in the cytoplasm, as well as the maintenance of the morpho/physiological characteristics for each cell line and their adhesion on the coverslips, with no detachment from the substrate, which could occur in case of cellular deterioration.

Both 10% *w*/*w* and 20% *w*/*w* show identical performance. Interestingly, cell adhesion appears not affected by the presence of the polymer. To confirm the absence of any cytotoxic effect, we further performed the hemocytometer counting assay, as reported in the Experimental Section. The histogram in Figure 5B shows the number of HaCaT, HUVEC, and BJ cells. Moreover, in this case, no notable differences could be proven. 

#### 3.3.2. Antimicrobial Assay

Two ubiquitous pathogenic bacteria were analyzed to investigate the ability of the P4VP-ZnNB film to mitigate or to prevent bacterial growth, i.e., the model strain *S. aureus* ATCC 29213 and the clinically isolated methicillin-resistant *S. aureus* WKZ-2. In both cases, the effects on bacterial growth were evaluated upon incubation of the bacteria with an uncoated P4VP film or with P4VP films functionalized with 10% and 20% ZnNB fragments. In each case, uncoated glass was tested as a control. In the two cases, we recorded a different but significant response, as shown in Figure 6.

Specifically, with the bacterial strain *S. aureus* ATCC, no significant effects on cell viability were detected in the case of uncoated glass. The neat P4VP showed no appreciable difference with respect to the uncoated glass, while significant antimicrobial activity was recorded in the case of the grafted polymers. In the case of the methicillin-resistant *S. aureus* MRSA WKZ-2 bacterial strain, the effects on cell viability were found to be similar in the presence of neat P4VP and the grafted polymers P4VP-ZnNB. Nevertheless, the effect was significant with respect to the uncoated glass, where the bacterial growth experienced no inhibition.

From these preliminary tests, it appears that the P4VP matrix represents a good choice for directing the antimicrobial activity. Remarkably, 20% samples showed a significant ability to interfere with bacterial growth, dependent on the specific features of target bacterial cells. Finally, the ability to mitigate the bacterial growth of a methicillin-resistant pathogen compared to a non-functionalized surface is an interesting result for a biomedical coating material. 

## 4. Conclusions

A novel PL hybrid material was obtained by grafting the ZnNB complex moiety in different percentages (10%, 20%, and 30% *w*/*w*) to a preformed P4VP polymeric chain. The best balance between PL performance and biological requirements was obtained with the 20% sample. A cross-analysis by diffractometric and DFT methods allowed the interpretation of the spectroscopic pattern of the coordination core before and after the grafting process. The coordination to the zinc (II) ion caused significant emission in the DR/NIR spectral region (PLQY = 35% in the solid state) and quite nominal stable grafting. The polymer P4PV-ZnNB 20% was easily formable and processable in emissive homogeneous transparent layers, highly adhesive, and stable in air and in a slightly acidic/basic water solution. Its biocompatibility in contact with the superficial layers of human skin was checked on human keratinocytes, endothelial cells, and fibroblasts, assessing the absence of a cytotoxic effect due to toxic release. The inhibition of the growth of operatory room pathogens such as *S. aureus* ATCC 29213 and *S. aureus* WKZ-2 was checked in a by-contact antimicrobial assay. The significant ability of P4VP-ZnNB 20% was recorded to interfere with bacterial growth, even for the methicillin-resistant pathogen, compared to a non-functionalized surface. These results suggest the possibility of employing P4VP-ZnNB 20% films as coatings for clinical devices. Based on the collected data, opportunities are envisaged for the design of stable, monitorable, and antimicrobial versatile hybrid tools.

## Data Availability

The data presented in this study are available on request from the corresponding author.

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
