# Peer review of "A DR/NIR Hybrid Polymeric Tool for Functional Bio-Coatings: Theoretical Study, Cytotoxicity, and Antimicrobial Activity"

_polymers, 2023, doi:10.3390/polym15040883_

Round 1

Reviewer 1 Report

A few points need to be seriously answered- check attachment

Author Response

Dear Editor,

Enclosed please find the revised version of our manuscript:

A DR/NIR hybrid polymeric tool for functional bio-coatings: theoretical study, cytotoxicity, and antimicrobial activity

By Rosita Diana, Francesco Silvio Gentile, Simona Concilio, Antonello Petrella, Raffaella Belvedere, Martina Schibeci, Angela Arciello, Luigi Di Costanzo, Barbara Panunzi

for publication in Polymers

We thank the reviewers for the constructive comments on the manuscript.

As indicated below, we have checked all the general and specific comments provided by the Referees and have made necessary changes accordingly to their indications.

Sincerely,

Prof. Barbara Panunzi

Reviewer #1

Here, I suggest some comments through the best of my knowledge on the Polymers manuscript for your consideration.

The manuscript by Rosita Diana, et al reports entitled “A DR/NIR hybrid polymeric tool for functional bio-coatings: theoretical study, cytotoxicity, antimicrobial activity”

  1. The author written ℃ as ◦C in many places (E.g. 106, 127, 143, 144), So follow the correct format in the manuscript and spaces from values to unit E.g. in 122 written 50°C (without space) but in 127 written as 330 °C (with space), follow the same format in the whole manuscript.

We thank the Reviewer; we emended the typos and minor corrections in all the manuscript.

  1. In synthesis of ZnNB section (2.2), why the author uses both the zinc (II) acetate and zinc (II) chloride salt for synthesis of ZnNB and the synthesis procedure using zinc (II) acetate was not mentioned, author mentioned only for zinc (II) chloride salt. From these 2 salts (zinc (II) acetate and zinc (II) chloride), which ZnNB compound gives best product result.

According with Reviewer’s observations, we clarified the concept in the text (lines 119-122)

  1. In section 2.2, the authors have not mentioned the elemental analysis (%) evaluation technique, mention the technique used to calculate the elemental analysis (%).

We understand the remark: in our case we relied upon an accredited external laboratory which performed the analyses according with standard methods.

  1. It is not mentioned, how the P4VP-ZnNB thin film were prepared. The author mentioned only the P4VP-ZnNB compound preparation. So, the author suggested to include the spin-coated procedure to prepare P4VP-ZnNB thin film.

In line 110-112 we did describe the spin-coating procedure. We added indication of solvent, concentration, and drying process in the same Pf.

  1. In section 2.1, the author mention the Mass spectrometry measurements were carried out, but the Mass spectrometry measurements was not seeming anywhere in the manuscript.

We are grateful for the observation. Mass spectrometry measurements was employed for ZnNB; we added the mass value in Pf. 2.2.

  1. In section 3.1, the author said that the P4VP-ZnNB 20% coordination fragment was fully grafted were shown by quite nominal zinc content. This indication is not enough for compound grafting result. By using other techniques given in manuscript, used to explain in detail to prove the P4VP-ZnNB 20% coordination fragment grafting. So, the author suggested to collect the preliminary IR data and include the 1H NMR spectrum, TGA, Elemental analysis explanation in Rand D section.

We thank the Reviewer. We already added NMR spectra in the Supplementary material. TGA analysis in hard copy has been added in the same section and the elemental analysis has been added in Pf 2.3. Crossed data from thermal and elemental analysis are in good agreement (also evidenced in lines 297-98).

  1. In Section 3.1, the author mentions 20% w/w ZnNB in P4VP shows better results than 10 and 30 %, but in section 3.3, the author mentions 10% sample was better. From 10% and 20% w/w P4VP-ZnNB, which % sample shows better result. Mention clearly.

Since we have looked for the best conditions from both biological activity and chemical-physical performance we can say that the best balance is given by the 20% polymer. It was clearly mentioned in lines 291-292: “As the best balance between PL performance and biological requirements was obtained with the 20% w/w ZnNB in P4VP (named P4VP-ZnNB 20%), it was fully characterised”. It was repeated in Conclusions, lines 451-452.

  1. In section 3.3.1, the author performed hemocytometer counting assay for in vitro compatibility of the polymer, the hemocytometer counting assay gives the preliminary understanding of biocompatibility, so it is not that much reliable assay for biocompatibility. Thus, this assay is not enough to proves the biocompatibility of the polymer coating. So, the authors will also suggest to include the additional MTT assay results for better biocompatibility performance. (MTT assay resulted in more accurate and reliable than hemocytometer counts)

We thank the Reviewer. First, we have modified Pf. 3.3.1 adding details about the

microscopic analysis. On the other hand, we must say that we could not perform the MTT assay. Indeed, the strong red colour of our samples, so evident in the bright field images, prevents the detection of the colorimetric variation due to alteration of the cell growth when compared to the control coverslips.

The manuscript is not giving the enough detail about the P4VP-ZnNB compound grafting, because the author mention it is novel material, so it is important to explains more about the compound, and that some further experiments should be performed. Before resubmitting, I strongly advise the authors to make the following changes.

Reviewer 2 Report

The manuscript is well written and organized.

I believe that it could be published in its present form.

It is not clear why Contribution/Funding sections are not completed.

It is stated that "These results suggest the possibility of employing P4VP-ZnNB 20% 454 films as coatings for clinical devices" it would be beneficial to the manuscript to mention about the stability of proposed polymeric material under medical sterilization conditions.

Author Response

Reviewer #2

The manuscript is well written and organized.

I believe that it could be published in its present form.

It is not clear why Contribution/Funding sections are not completed.

We thank the Reviewer, Contribution/Funding sections have been completed.

It is stated that "These results suggest the possibility of employing P4VP-ZnNB 20% 454 films as coatings for clinical devices" it would be beneficial to the manuscript to mention about the stability of proposed polymeric material under medical sterilization conditions.

The films used in all biological assays were sterilised through UV rays. According with Reviewer’s observations, in Pf. 2.1 we added information about the sterilization conditions used, which were not harmful to the film.
